# Flexible Conditional Generation with Stochastically Factorized Autoregressive Models

## Abstract

Deep autoregressive generative models have demonstrated promising results in unconditional generation tasks for structured data, such as images. However, their effectiveness in conditional generation remains relatively underexplored. We hypothesize that previous models, including autoregressive diffusion models, need to efficiently amortize across all possible generation orders, allowing them to parameterize any conditional distribution between data elements. This capability makes them particularly well-suited for data acquisition tasks, where collecting data to maximize predictive accuracy is critical. In this work, we propose a novel method for active data acquisition using autoregressive diffusion models, achieving promising results. However, we observe that when these models are trained with a fixed deterministic or stochastic order, they often struggle to generate content accurately during testing if the masking mechanism changes. To address this, we introduce a new deep generative model that leverages intrinsic data information along with self-supervision principles. Our approach is both simple and highly flexible, extending established autoregressive frameworks by probabilistically modeling the per-element generation process as a mixture of semi-supervised mechanisms. This method provides a robust framework for conditional generation across various masking patterns. Experimental results demonstrate significant improvements in both simplicity and accuracy for conditional generation tasks, outperforming conventional methods that rely on random permutations or simultaneous generation of all dimensions.

## 1 Introduction

The ability to effectively generate data conditioned on prior inputs or specific attributes can significantly enhance the relevance and utility of model outputs in practical settings. Conditional generation tasks are crucial for developing generative models that can produce specific outcomes based on given conditions, a capability essential for applications ranging from personalized content creation to targeted data augmentation. Among the various deep generative models, latent variable models such as Variational Autoencoders (VAEs, Kingma & Welling, 2013) or Denoising-Diffusion Probabilistic Models (DDPMs, Ho et al., 2020) offer efficient frameworks for these tasks by modelling the joint distribution of data and conditions and generating data from the conditioned latent space. VAEs can efficiently handle incomplete data (Mattei & Frellsen, 2019; Nazabal et al., 2020; Ma et al., 2020; Peis et al., 2022). However, they require approximate inference, selecting appropriate priors (Maaløe et al., 2019; Vahdat & Kautz, 2020; Child, 2020; Tomczak & Welling, 2018), and inference models (Kingma et al., 2016; 2021) can introduce significant complications. DDPMs have shown promise in matching conditional distributions accurately but often produce predictive outcomes that do not align semantically with the conditioning inputs (Lugmayr et al., 2022). Adversarial methods like Generative Adversarial Networks (GANs, Goodfellow et al., 2014) provide powerful capabilities for generating high-quality images but are notoriously difficult to train and, in their basic form, do not naturally support conditional generation.

In contrast, deep autoregressive models are particularly well-suited for conditional generation tasks, thanks to their inherent design and straightforward learning process via maximum likelihood estimation. In the last decade, they have emerged as state-of-the-art in the generative model landscape (Graves, 2013; Van den

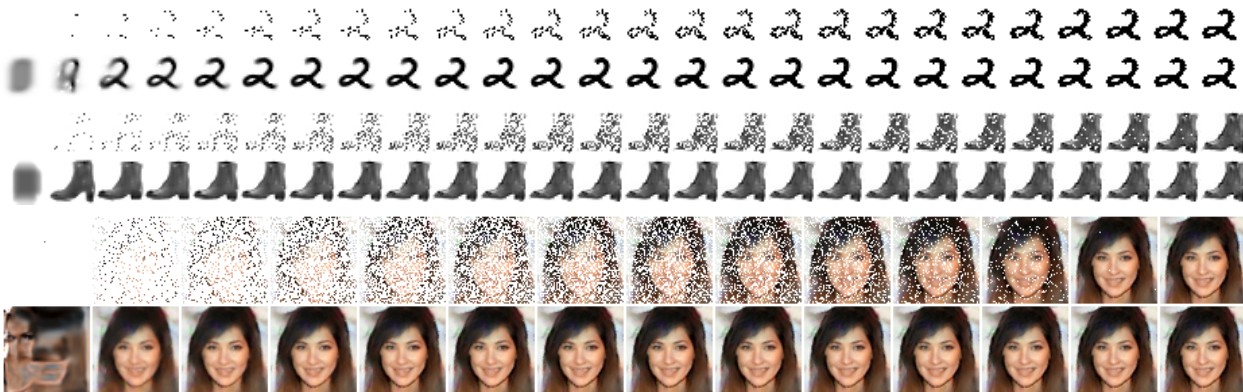

Figure 1: Samples generated by `SFARM` trained with different datasets. Columns represent intermediate generation steps. Odd rows represent the sampled $\boldsymbol{x}_{\boldsymbol{z}_i}$. Even rows show the likelihood parameters given by $f_\theta(\boldsymbol{x}_{\Delta\boldsymbol{z}_i}|\boldsymbol{x}_{\boldsymbol{z}_{i-1}})$.

Oord et al., 2016c;b;a; Brown et al., 2020). While these models excel in scenarios where the order of data points is critical, adapting them for applications without a clear natural ordering, such as pixel generation within images, remains challenging. Conventional methods typically restrict generation to fixed sequences, such as left-to-right or top-to-bottom (Van den Oord et al., 2016c;b; Salimans et al., 2017), which limits the flexibility required to synthesize complex, diverse, and high-dimensional datasets.

Apart from including auxiliary models to provide abstract information (Van den Oord et al., 2016b), various strategies have been proposed to circumvent these constraints. Order-agnostic autoregressive models (Uria et al., 2014; Hoogeboom et al., 2022) encompass frameworks for amortizing the unsorted generation by accumulating elements with random order permutations and feeding them to a single model for predicting new elements. They show success in efficiently learning the $\mathcal{O}(S!)$ possible generation trajectories. In our work, we propose a wider family of models, for which we show that Autoregressive Diffusion Models (ARDMs, Hoogeboom et al., 2022) can be reformulated as a specific case. We design the order mechanism by pre-defining Markov chain transition probabilities, with one possible choice being the discrete uniform, equivalent to a random permutation of the elements.

Our main contributions are the following:

- We present a new family of order-agnostic autoregressive models that generalizes previous approaches by defining a mixture of probabilistic activation mechanisms via Markov transition probabilities for generating new elements.

- We show that previous order-agnostic models with amortization over random permutations can be reformulated within our definitions.

- We demonstrate the effectiveness of our proposed method in efficient information acquisition.

- We demonstrate the superiority of our proposed family of models over alternatives in relevant downstream tasks that require conditional generation.

## 2  Related Work

**Autoregressive Models**  Autoregressive models generate data by following a predefined factorization order. While this order is intuitive for time series or video data, it is less apparent for images. PixelRNN (Van den Oord et al., 2016c) introduced a method for sequentially constructing images from top to bottom and left to right using recurrent neural networks. PixelCNN (Van den Oord et al., 2016b) later adopted a convolutional approach to pixel-by-pixel image generation, improving computational efficiency and scalability

while retaining high-quality synthesis. PixelCNN++ (Salimans et al., 2017) further advanced PixelCNN by incorporating gated convolutional layers, downsampling for larger images, and a discretized logistic mixture likelihood function to enhance output quality.

Despite these advancements, PixelCNN and its variants struggled to generate diverse and realistic samples from complex datasets. To address this, the authors conditioned the generation process on a pre-trained classifier's output from CIFAR-10 and integrated PixelCNN as a decoder within a Variational Autoencoder (VAE) framework, leading to the development of PixelVAE (Gulrajani et al., 2016). Nevertheless, PixelCNN's architecture still faced limitations in independently producing a wide array of complex, realistic images.

Recent advancements in autoregressive models and Latent Diffusion Models (Rombach et al., 2022) have inspired new approaches, such as training autoregressive models on learned latent spaces. For example, Kingma et al. (2016) demonstrated how inverse autoregressive flow could enhance the flexibility of the variational posterior. Similarly, Tschannen et al. (2023) explored fitting Transformers to the latent space of a $\beta$-VAE (Higgins et al., 2017; Burgess et al., 2018), while Dupont et al. (2022) attempted to fit Autoregressive Transformers to modulation vectors representing data functions. However, challenges in establishing a meaningful ordering have hindered the success of these approaches.

**Autoregressive Diffusion Models**  The flexibility of autoregressive models can enhanced by amortizing the generation of any element at any step with a single network. Building on the order-agnostic training procedure proposed by Uria et al. (2014) and the efficient learning process of DDPMs (Ho et al., 2020), Hoogeboom et al. (2022) uniformly sample the space of permutations of pixel indices and parameterized the resulting conditional likelihoods via U-Nets (Ronneberger et al., 2015). Their method, ARDM, which is equivalent to Discrete Diffusion Models when using the absorbing state (Austin et al., 2021), is also applicable to continuous data, and demonstrated remarkable generation quality and was successfully applied to lossless compression. Our approach aligns with ARDM's learning objective but generalizes it by explicitly modelling the activation probability per step and element as a Markov chain. This formulation accommodates any stochastic factorization, with random permutation being just one possible mechanism. We will elaborate on this in the subsequent sections.

**Semi-supervision within autoencoding frameworks**  The application of semi-supervised techniques in VAEs, particularly through the use of masked data (He et al., 2022), has revolutionized the field of deep generative models by introducing new strategies for improving interpretability (Senetaire et al., 2023), or incomplete data handling by uniformly masking training data (Ma et al., 2019; 2020; Peis et al., 2022) and factorizing the likelihood across elements (Mattei & Frellsen, 2019; Nazabal et al., 2020). These approaches facilitate robust conditional generation, enabling the models to infer unobserved data segments based on the observed portions.

Other examples of introducing semi-supervision to VAEs include the *Self-VAE* (Gatopoulos & Tomczak, 2021), which utilizes deterministic and discrete transformations of data for both conditional and unconditional sampling while simplifying the objective function. The study explores the use of single self-supervised transformations like downscaling or edge detection as latent variables and extends this concept to a hierarchical architecture, demonstrating benefits over traditional VAEs. The *Super-Resolution VAE* (Gatopoulos et al., 2020) addresses the issue of blurriness in generated images, a common drawback in traditional VAEs, by incorporating semi-supervision: adding a downscaled version of the original image as a random variable in the model.

## 3  Stochastically Factorized Autoregressive Models

In this section, we present our proposed method. We introduce `SFARM` in the context of autoregressive modelling to later derive its likelihood-based objective and its simplification for efficient learning.

### 3.1 Notation

Let us denote a data point by $\boldsymbol{x} \in \mathcal{X}$, where $\mathcal{X}$ represents the data space. For example, in tabular data, $\mathcal{X}$ is in $\mathbb{R}^D$, with $D$ the dimensionality or number of features, and in the context of images, $\mathcal{X}$ is $\mathbb{R}^{C \times H \times W}$, being $C$ the number of channels, $H$ and $W$ the height and width, respectively. We use $j$ for indexing the sorted *elements* of a data point, such as a pixel within an image. Whilst the number of elements in tabular data corresponds to the dimensionality itself of a data point, an image consists of $H \times W$ elements, denoted as pixels, where each is of dimensionality $C$. Importantly, our $i$ refers to the generation step, which does not follow any spatial or temporal natural orders.

### 3.2 Introducing `SFARM`

Within an autoregressive framework, $\boldsymbol{x}_i$ is denoted as each intermediate stage of a generated data point, culminating in the final observable $\boldsymbol{x}$. We consider the autoregressive probability distribution

$$p(\boldsymbol{x}) = \prod_{i=1}^{S} p(\boldsymbol{x}_i | \boldsymbol{x}_{<i}). \tag{1}$$

Equation equation 1 is generally applied to a pre-defined deterministic generation trajectory. In broader terms, one can decide if features in tabular data are generated left-to-right, or if pixels of an image are generated sequentially in a left-right, top-bottom order, like in Van den Oord et al. (2016c;b); Salimans et al. (2017); Gulrajani et al. (2016). These constraints limit the expressivity of a network for synthesizing complex high-dimensional data. As demonstrated by Van den Oord et al. (2016b), auxiliary models are required to guide the generation process and increase the diversity.

Our proposed model, depicted in Figure 2 and denoted as `SFARM` for Stochastically Factorized AutoRegressive Model, distinguishes itself from previous autoregressive models for images, such as (Van den Oord et al., 2016c;b; Salimans et al., 2017), by allowing the intermediate stages of the autoregressive generation, denoted by $\boldsymbol{x}_{\boldsymbol{z}_i}$, to follow a stochastic schedule. This is achieved by indexing $\boldsymbol{x}$ with a latent variable $\boldsymbol{z}_i$, rather than a deterministic one. We make use of a random *activation mask*, a binary variable with the same structure of the data, denoted by $\boldsymbol{z}$. For tabular data, $\boldsymbol{z} \in \{0,1\}^D$, whilst for images, $\boldsymbol{z} \in \{0,1\}^{H \times W}$. Specifically, the notation $\boldsymbol{x}_{\boldsymbol{z}_i}$ represents the data elements in $\boldsymbol{x}$ indexed by $\boldsymbol{z}_i$.

In simpler terms, each structured element of $\boldsymbol{z}_i$ is a binary indicator determining whether the corresponding element in $\boldsymbol{x}$ is active. An element being *inactive* means it is zero-masked in $\boldsymbol{z}_i$. A collection of sequential activation masks $\boldsymbol{z}_{0:S}$, where $S$ refers to the number of generation steps (i.e., $S \equiv D$ for tabular data, $S \equiv H \times W$ assuming only one dimension is revealed at every step), defines the order or autoregressive generations.

To further increase the generation flexibility, we assume that the sequence of masks comes from a uniform mixture of masking mechanisms, where we can easily define different masking schemes to train the model for conditional generation,

$$p(\boldsymbol{z}_{0:S}) = \frac{1}{M} \sum_{k=1}^{M} p(\boldsymbol{z}_{0:S} | m). \tag{2}$$

Within these definitions, the joint distribution of a sequence of masks, a masking mode, and an observed data point can be expressed as

$$p(\boldsymbol{x}, \boldsymbol{z}_{0:S}, m) = p(\boldsymbol{z}_0) \frac{1}{M} \prod_{i=1}^{S} p(\boldsymbol{z}_i | \boldsymbol{z}_{i-1}, m) \, p_\theta(\boldsymbol{x}_{\Delta \boldsymbol{z}_i} | \boldsymbol{x}_{\boldsymbol{z}_{i-1}}, i), \tag{3}$$

where the notation $\Delta \boldsymbol{z}_i$ refers to the difference between activation masks $\boldsymbol{z}_i$ and $\boldsymbol{z}_{i-1}$, i.e., the points activated at step $i$.

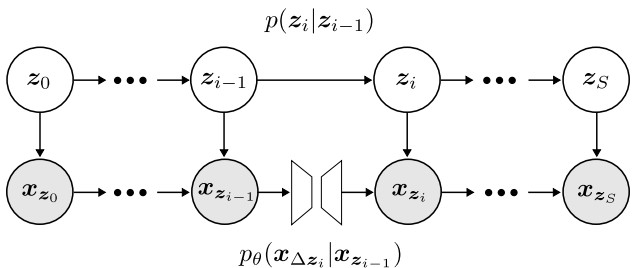

Figure 2: Probabilistic graph of the `SFARM` generative model.

### 3.3 Stochastic Factorization

As stated above, we can probabilistically model the patterns ruling the activation of new elements, by defining $p(\boldsymbol{z}_i|\boldsymbol{z}_{i-1}, m)$ as a first-order Markov chain. In the following lines, we are omitting the conditioning on $m$ for simplicity. The dynamics of this chain are detailed by two conditionally independent distributions:

$$p(\boldsymbol{z}_i|\boldsymbol{z}_{i-1}) = p(\boldsymbol{z}_i^\uparrow|\boldsymbol{z}_{i-1}) \cdot p(\boldsymbol{z}_i^\downarrow|\boldsymbol{z}_{i-1}). \tag{4}$$

Here, $\boldsymbol{z}_i^\uparrow$ refers to elements already active in $\boldsymbol{z}_{i-1}$ ("up"), with each of these elements following a delta distribution, $p(z_{ij}^\uparrow|\boldsymbol{z}_{i-1}) = \delta(1)$ that enforces them to remain activated. Conversely, $\boldsymbol{z}_i^\downarrow$ includes elements yet to be activated ("down"), i.e. $\boldsymbol{z}_i^\downarrow = (z_{ik} : z_{i-1,k} = 0)$. It is the distribution of this variable that will define the activation process. Within our framework, any discrete activation distribution can be chosen, with the only constraints of $p(\boldsymbol{z}_0) = \delta(\mathbf{0})$, denoting the empty mask, and conversely $p(\boldsymbol{z}_S) = \delta(\mathbf{1})$, referring to the fully activated mask. In particular, we allow for both generating single or multiple elements within each step.

### 3.4 `SFARM` likelihood-based objective

In our approach, we conceptualize the proposed method as a latent variable model, with the latent variable being a sequence of binary matrices representing a trajectory of element activations. The Markov transition probabilities of the latent sequence are pre-defined by design.

The goal is to amortize the generation of any new element in $\boldsymbol{z}_i$, which we denote by $\Delta\boldsymbol{z}_i$, activated under the stochastic schedule. With that purpose, we define a neural imputation model $p_\theta(\boldsymbol{x}_{\Delta\boldsymbol{z}_i}|\boldsymbol{x}_{\boldsymbol{z}_{i-1}}, i, m)$ implemented by a U-Net network (Ronneberger et al., 2015), with learnable parameters $\theta$ for predicting the next data element from the previous state, and embeddings of both the generation step $i$, and the masking mode $m$. Like in DDPMs (Ho et al., 2020), time embeddings of the step $i$ are fed into this network.

The log marginal likelihood can be expressed as

$$\log p(\boldsymbol{x}) = \log \mathbb{E}_{p(\boldsymbol{z}_{0:S}, m)}[p_\theta(\boldsymbol{x}|\boldsymbol{z}_{0:S}, m)] \geq \mathbb{E}_{p(\boldsymbol{z}_{0:S}, m)}\left[\log p_\theta(\boldsymbol{x}|\boldsymbol{z}_{0:S}, m)\right] = \mathcal{L}(\boldsymbol{x}; \theta), \tag{5}$$

which holds by Jensen's inequality and defines a proper Evidence Lower Bound (ELBO). Note that our lower bound is equivalent to the standard ELBO used in VAEs if the variational posterior $q(\boldsymbol{z}|\boldsymbol{x})$ coincides with the prior $p(\boldsymbol{z})$. As in a diffusion-based model, but with a different mechanism, we give up learning any meaningful structure in the latent space for the sake of flexible autoregressive generation. The parameters $\theta$ are thus learned by performing SGD for the optimization problem

$$\theta^* = \arg\max_\theta \left\{ \sum_n \mathcal{L}^{(n)}(\boldsymbol{x}^{(n)}; \theta) \right\}, \tag{6}$$

where, based on the autoregressive properties, we can express the log-likelihood from equation 5 as

$$\mathcal{L}(\boldsymbol{x}, \theta) = \sum_{i=0}^{S} \mathbb{E}_{p(\boldsymbol{z}_{0:i}, m)}\left[\log p_\theta(\boldsymbol{x}_{\Delta\boldsymbol{z}_i}|\boldsymbol{x}_{\boldsymbol{z}_{i-1}}, i, m)\right] = \sum_{i=0}^{S} \mathcal{L}_i(\boldsymbol{x}, \theta). \tag{7}$$

---

**Algorithm 1** Training `SFARM`

---

**Input:** $\boldsymbol{x}^{(1:N)}$, $f_\theta$, $p(\boldsymbol{z}_{0:S})$
**repeat**
    $i \sim U(0,S)$,   $m \sim U(0,m)$ $\boldsymbol{z}_{i-1} \sim p(\boldsymbol{z}_{i-1})$
    $\hat{\mathcal{L}}_i = \frac{1}{S-i+1} \sum \log p_\theta(\boldsymbol{x}_{\boldsymbol{z}_i^\downarrow}|\boldsymbol{x}_{\boldsymbol{z}_{i-1}})$
    $\theta \leftarrow \text{Adam}_\theta(\hat{\mathcal{L}}_i)$
**until** converged

---

**Algorithm 2** Sampling from `SFARM`

---

**Input:** $f_\theta$, $p(\boldsymbol{z}_{0:S})$
**Initialize:** $\boldsymbol{x}_{\boldsymbol{z}_0} = \boldsymbol{0}$
**for** $i = 1, \ldots, S$ **do**
    $\boldsymbol{x} \sim p_\theta(\boldsymbol{x}|\boldsymbol{x}_{\boldsymbol{z}_{i-1}})$
    $\boldsymbol{z}_i \sim p(\boldsymbol{z}_i|\boldsymbol{z}_{i-1})$
    $\boldsymbol{x}_{\boldsymbol{z}_i} = \boldsymbol{x}_{\boldsymbol{z}_{i-1}} + \boldsymbol{x}_{\Delta \boldsymbol{z}_i}$
**end for**

---

Here, we employ the notation $\mathbb{E}_{p(\boldsymbol{z}_{0:i})}$ for a nested expectation $\mathbb{E}_{p(\boldsymbol{z}_0)}\left[\mathbb{E}_{p(\boldsymbol{z}_1|\boldsymbol{z}_0)}[...]\right]$, and we define $\mathcal{L}_i(\boldsymbol{x}, \theta)$ as the $i$-th term of the ELBO.

To compute the ELBO, as defined in Eq. equation 7, one must consider the summation over all possible trajectories and masking modes that could lead to generating a data point. Specifically, this involves exploring all the $\mathcal{O}(S!)$ number of trajectories, or equivalently, all the conditional models $p(\boldsymbol{x}_{\boldsymbol{z}_i}|\boldsymbol{x}_{\boldsymbol{z}_{i-1}})$ defined by any $\boldsymbol{z}_{:S}$ under any stochastic activation scheme, $m$. However, even for data structures with a few elements, the count of possible activation sequences becomes prohibitively large.

Consequently, an approximation of the ELBO is required. A straightforward approach would involve (i) approximating the objective via Monte Carlo methods by sampling entire sequences from $p(\boldsymbol{z}_{0:S})$ and (ii) processing the entire trajectory to perform a single optimization step. However, this poses a significant computational challenge, especially for datasets with a large number of elements. To address this, we propose an alternative, simplified objective that significantly enhances the computational efficiency of the learning process.

### 3.4.1 Simplified objective

In addressing the computational challenges inherent in autoregressive models, particularly with high-dimensional data, we propose an alternative formulation of the objective in equation 7,

$$\mathcal{L}(\boldsymbol{x}; \theta) = S \cdot \mathbb{E}_{\mathcal{U}(i)}\left[\mathcal{L}_i(\boldsymbol{x}; \theta)\right], \tag{8}$$

where drawing inspiration from DDPMs' (Ho et al., 2020) amortization over time steps, we use the Law of the Unconscious Statisticians (LOTUS) to reformulate the summation over generation steps as an expectation over uniformly distributed steps $i$. By drawing a sample $i \sim \mathcal{U}(0, S)$, we approximate the ELBO by a single term, i.e., $\hat{\mathcal{L}}(\boldsymbol{x}; \theta) = S \cdot \mathcal{L}_i(\boldsymbol{x}; \theta)$. Further, by utilizing the Law of Total Expectation, $\mathbb{E}_{p(\boldsymbol{z}_{i-1})}\left[\mathbb{E}_{p(\boldsymbol{z}_i|\boldsymbol{z}_{i-1})}[f(\boldsymbol{z}_i)]\right] = \mathbb{E}_{p(\boldsymbol{z}_i)}[f(\boldsymbol{z}_i)]$, a more compact expression of $\mathcal{L}_i$ can be formulated. This law simplifies the nested expectation of each term in equation 7 to

$$\mathcal{L}_i(\boldsymbol{x}, \theta) = \mathbb{E}_{p(\boldsymbol{z}_{i-1}, m)}\left[\mathbb{E}_{p(\boldsymbol{z}_i|\boldsymbol{z}_{i-1}, m)}\left[\log p_\theta(\boldsymbol{x}_{\Delta \boldsymbol{z}_i}|\boldsymbol{x}_{\boldsymbol{z}_{i-1}}, i, m)\right]\right], \tag{9}$$

enabling us to articulate the objective in terms of marginal probabilities at step $i-1$. The outer expectation can be approximated via Monte Carlo if we have access to samples from the marginal distribution of the mask at any step $p(\boldsymbol{z}_i)$. The inner expectation can be analytically solved and gives

$$\hat{\mathcal{L}}_i(\boldsymbol{x}, \theta) = \sum_{z_i \in \boldsymbol{z}_i^\downarrow} p(z_i|\boldsymbol{z}_{i-1}, m) \log p_\theta(\boldsymbol{x}_{z_i}|\boldsymbol{x}_{\boldsymbol{z}_{i-1}}, i, m). \tag{10}$$

This approach enhances computational efficiency while maintaining the model's effectiveness in learning the desired element activation trajectories. In simpler terms, for images, we focus on training to impute unobserved pixel portions of images using randomly incomplete versions as inputs, and weight the per-element likelihoods according to the pre-defined kernels. Notably, our method stands out as i) it does not require an additional regularization term, simplifying the training process while maintaining robust and effective performance; and ii) it permits performing optimization steps with a single forward pass, and without necessarily forwarding through the entire trajectory.

**Special case of uniform activation distribution** We can easily impose a one-element-per-step genera-tion by defining the activation distribution as Categorical. By setting the initial activation to $p(z_{0j}) = \delta(0)$ for all elements, i.e., no active data points, complete data observation is ensured at $i = S$. Furthermore, we can restrict to uniform activation, meaning that every non-activated pixel is equally likely to be activated, by choosing a uniform Categorical distribution with probability $\frac{1}{S-i+1}$. With these assumptions, and con-sidering the case with a single masking mode, equation 10 coincides with the learning objective of ARDMs (Hoogeboom et al., 2022),

$$\hat{\mathcal{L}}_i(\boldsymbol{x}, \theta) = \frac{1}{S-i+1} \sum_{z_i \in \boldsymbol{z}_i^{\downarrow}} \log p_\theta(\boldsymbol{x}_{z_{ik}} | \boldsymbol{x}_{\boldsymbol{z}_{i-1}}, i). \tag{11}$$

which is also a widely spread strategy in previous successful autoregressive models for text like BERT (Devlin, 2018), or for images, like MaskGIT (Chang et al., 2022), that use uniform masking schemes. Additionally, we derive the marginal distribution of elements at any given step using the total probability theorem (details are in Appendix B.1), ending in $p(\boldsymbol{z}_i) = \mathcal{C}(\alpha_i)$ with $\alpha_i = \frac{i}{S}$. This parameter indicates the marginal probability of an element's activation at step $i$, reflecting the proportion of $i$ relative to the total number of steps $S$. In fact, as we will demonstrate later, this choice allows for encouraging generation diversity by evenly weighting all the amortized autoregressive models.

### 3.5 Masking Modes

Our proposed `SFARM` offers greater flexibility compared to alternative methods such as ARDMs, primarily due to its customizable activation sequences. By carefully designing $p(\mathbf{z}_{0:S})$, `SFARM` can better accommodate various meaningful activation mechanisms, which is particularly beneficial for conditional generation tasks. For instance, when applied to images, generating interior patches based on existing content can be highly useful. However, restricting the model to a single generation mechanism may not be suitable for all scenarios.

To address this limitation, we propose that $\mathbf{z}_{0:S}$ be derived from a **mixture of activation mechanisms**, allowing us to incorporate multiple activation strategies as required. We define $m$ as the masking mode indicator, enabling us to express $p(\mathbf{z}_{0:S}|m)$, as shown in equation 2.

For image-based tasks, we consider up to seven activation mechanisms, including random, left-to-right, right-to-left, top-to-bottom, bottom-to-top, interior patches, and exterior patches. These modes are defined by dividing the activation sequence into two stages, $(a)$ and $(b)$, where complementary spatial regions are generated in each stage, and the second stage in conditioned on the first. This partitioning facilitates a consistent mathematical formulation for the distributions, irrespective of the masking mode $m$:

$$p(\boldsymbol{z}_{0:S}|m) = p(\boldsymbol{z}_{0:S'}^{(a)}|m) \cdot p(\boldsymbol{z}_{S':S}^{(b)}|\boldsymbol{z}_{0:S'}^{(a)}, m). \tag{12}$$

Within this framework, $p(\mathbf{z}_{0:S'}^{(a)})$ and $p(\mathbf{z}_{S':S}^{(b)}|\mathbf{z}_{0:S'}^{(a)})$ are defined similarly to the previously introduced $p(\mathbf{z}_{0:S})$. New points are activated uniformly using a categorical distribution over the non-activated points within stages $(a)$ and $(b)$. The distinction between $(a)$ and $(b)$ lies in their respective spatial regions, each containing a distinct set of elements, allowing for more diverse and flexible generation patterns.

### 3.6 Likelihood Approximation

For training, we use a simplified objective that leads to better results and faster convergence. However, to evaluate the model, we seek more accurate approximations of the true likelihood. Given that the sample space of $\boldsymbol{z}$ is finite and discrete, the marginal likelihood can be written as

$$p(\boldsymbol{x}) = \mathbb{E}_{p(\boldsymbol{z})}[p_\theta(\boldsymbol{x}|\boldsymbol{z})] = \sum_{k=1}^{S!} p_\theta(\boldsymbol{x}|\boldsymbol{z}^{(k)}) p(\boldsymbol{z}^{(k)}). \tag{13}$$

Here, $\boldsymbol{z}$ represents the full activation sequence, and we omit the sub-index $_{0:S}$ for simplicity. Each $\boldsymbol{z}^{(k)}$ corresponds to one of the $S!$ possible activation sequences. While this expression is analytically solvable,

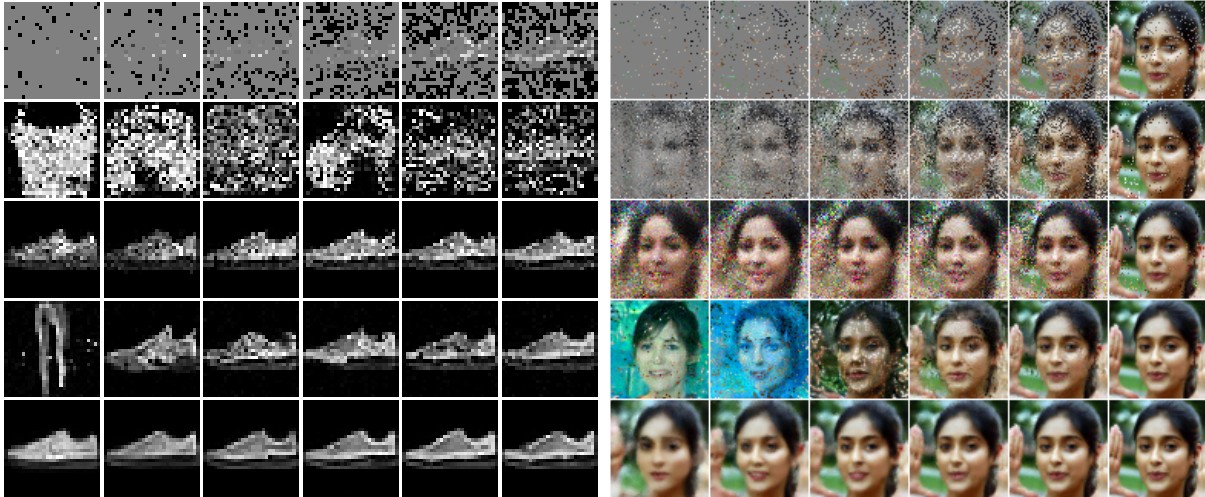

Figure 3: Image imputation with different models and datasets. First row: incomplete data fed to the model. Second to last row: data completion by VAE, MIWAE, RePaint and `SFARM`. Columns represent unobserved percentages in [5,10,20,30,50,90].

considering all $S!$ activation sequences is impractical. A Monte Carlo approximation,

$$p(\boldsymbol{x}) \approx \frac{1}{K} \sum_k p_\theta(\boldsymbol{x}|\boldsymbol{z}^{(k)}), \tag{14}$$

would be particularly effective if $p(\boldsymbol{z}_i^\downarrow|\boldsymbol{z}_{i-1})$ is set as a discrete uniform, and for this reason we choose to fix the masking mode to uniform for evaluating the likelihood. This approach, unlike other types of models, avoids the need for more sophisticated approximations, e.g., importance weighting (Mattei & Frellsen, 2019) or MCMC methods (Caterini et al., 2018; Thin et al., 2021) in VAEs.

### 3.7 Efficient data acquisition with `SFARM`

Learning conditional distributions is particularly relevant when active learning downstream tasks are considered. Active Feature Acquisition (Melville et al., 2004; Saar-Tsechansky et al., 2009; Huang et al., 2018) stands out among various Active Learning strategies, particularly in cost-sensitive scenarios where it is crucial to balance the enhancement of predictive accuracy with the expenses associated with obtaining new feature-level data. Recent contributions have explored deep generative modelling frameworks for iteratively gathering valuable insights by selecting features that optimize an information-theoretic reward function, thereby improving prediction quality. The reward function can be defined within `SFARM` notation as

$$R(i, \Delta \boldsymbol{x}_{\boldsymbol{z}_i}) = \mathcal{I}(\boldsymbol{x}_\Phi; \boldsymbol{x}_{\Delta \boldsymbol{z}_i}|\boldsymbol{x}_{\boldsymbol{z}_{i-1}}), \tag{15}$$

where $\boldsymbol{x}_\Phi$ is the target element. This process is known as Sequential Active Information Acquisition (SAIA). Ma et al. (2019; 2020) proposed practical approaches for estimating a complex reward function using a VAE encoder that accommodates incomplete data. Peis et al. (2022) further developed the framework using a simpler sampling-based method for estimating the reward function, i.e. the Mutual Information.

However, all of the VAE-based methods require intractable marginalizations of the continuous latent variables in order to compute the reward in equation 15, and mainly opt by approximations based on Gaussian proposals. In contrast with VAEs, we have access to the conditional distribution of any subset of the data point structure given the rest, without needing to approximate an intractable integral.

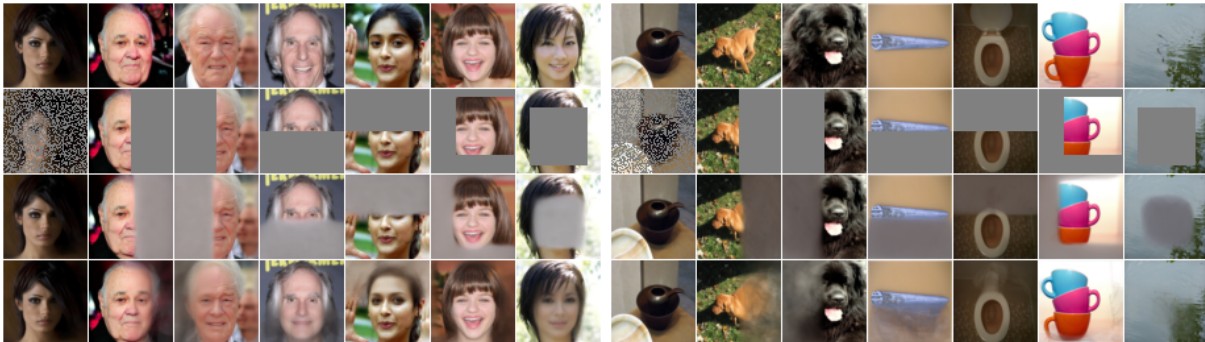

Figure 4: Conditional generation by ARDM (Hoogeboom et al., 2022) (third row) and our `SFARM` (last row). First row: original test data. Second row: incomplete data fed to the model.

## 4 Experiments

The evaluation of `SFARM` is organized into quantitative and qualitative experiments that empirically analyze its performance. Namely, we compare our method with the following deep generative models that allow for a flexible and efficient conditional generation and use similar neural architectures:

- **VAE** from (Kingma & Welling, 2013) trained with fully observed data.
- **MIWAE**: an Importance Weighted Autoencoder (Burda et al., 2016) variation from Mattei & Frellsen (2019) that handles incomplete data more accurately.
- **DDPM**: a Denoising Diffusion Probabilistic Model (Ho et al., 2020). For conditional generation, we use the RePaint strategy presented by Lugmayr et al. (2022).

We employ the same proposed network architecture for every model. All the autoencoding-based models use the top-down part of the network as encoder and a bottom-up part as the decoder, without including skip connections. All the details on the experimental setup and reproducibility can be found in A. We evaluate `SFARM` over diverse image benchmark datasets, including MNIST, Fashion-MNIST, CIFAR10, CelebA-HQ (64,64) and ImageNet (64, 64), and tabular datasets, including Energy, Yatch and Concrete. Our code is accessible at `https://anonymous.4open.science/r/FARM_pre-D61C/`.

### 4.1 Conditional generation

In this section, we evaluate the conditional generation capabilities of `SFARM` and compare it with baseline models. Figure 3 shows the results for image completion when the input is uniformly masked. The columns represent the percentage of non-activated (missing) pixels, i.e., $\mathbb{E}[z_i]$ for the uniform case. The first row displays the input to the model, rows 2–4 show the outputs from the baselines, and row 5 presents the outputs from `SFARM`.

Table 1: Test NLL of tabular datasets on the normalized unobserved features by our model and baselines.

|  | VAE | MIWAE | SFARM |
|---|---|---|---|
| Energy | $4.90 \pm 0.59$ | $4.74 \pm 0.56$ | $\mathbf{0.21 \pm 2.14}$ |
| Yatch | $4.42 \pm 0.83$ | $4.14 \pm 1.22$ | $\mathbf{2.50 \pm 1.19}$ |
| Concrete | $4.10 \pm 0.08$ | $3.06 \pm 0.17$ | $\mathbf{1.05 \pm 0.06}$ |

We highlight the ability of `SFARM` to fill in the non-observed pixels compared to the baselines, even when only 5% of the image is observed. Except for MIWAE and `SFARM`, other models struggle to achieve accurate details when larger portions of the image are missing. `SFARM` inherently processes global properties in fewer steps and incorporates finer details earlier than MIWAE, leading to more accurate and realistic image completions.

Next, we evaluate the performance of `SFARM` compared to ARDM, specifically in the context of using a uniform masking mode. As shown in Figure 4, ARDM has difficulty conditionally generating content when the observed mask differs from the uniform mask seen during training. In contrast, `SFARM` maintains robustness across varying masking patterns, demonstrating its superior ability to adapt to different conditions.

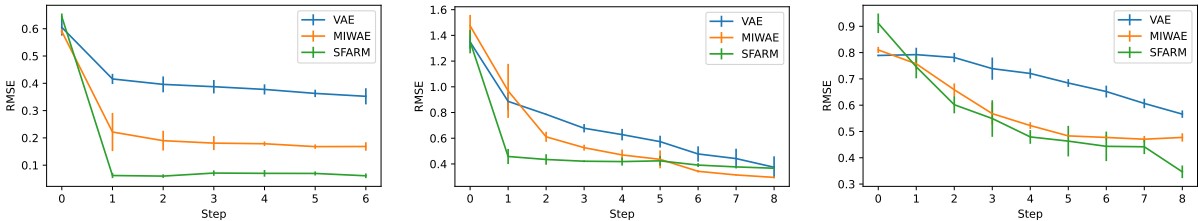

Figure 5: SAIA experiment for the Yatch (a), Energy (b) and Concrete (c) datasets.

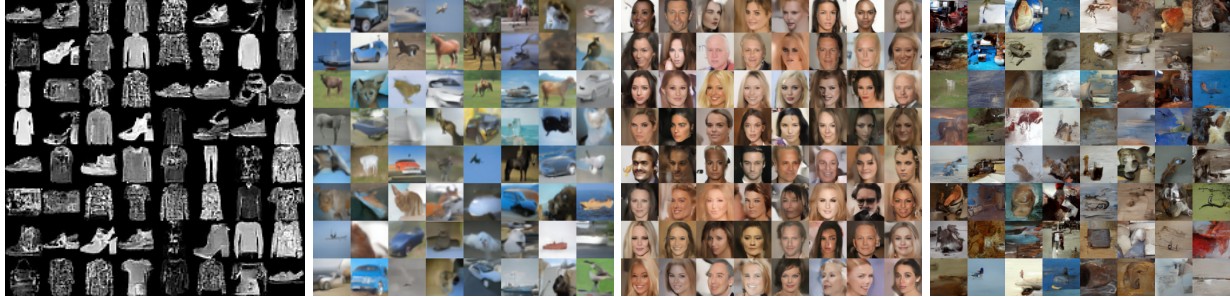

Figure 6: Samples from `SFARM` trained on Fashion-MNIST, CIFAR-10, CelebA-HQ (64,64) and ImageNet (64,64). We employ temperature parameters to shrink the scale of the learned distributions, which increases the quality of the images with the sacrifice of decreasing diversity.

To demonstrate the flexibility of `SFARM` in handling various data structures, we include results on missing data imputation with tabular datasets from UCI (Dheeru & Taniskidou, 2017), specifically on Energy, Yatch and Concrete. In this context, our model can be seen as a mixture of all possible belief networks with sequential knowledge. We employ simple, 3-layered MLPs, and concatenate time embeddings to the input, for modelling the conditional distributions. The results reported in Table 1 show that `SFARM` is superior over the baselines.

## 4.2 Active information acquisition

Results for the SAIA scenario are presented in Figure 5. It is evident that foundational VAEs do not offer an accurate solution for dynamically acquiring new variables. The recent MIWAE model demonstrates increased accuracy and faster discovery of relevant information. As it can be appreciated, our proposed method outperforms MIWAE, indicating that `SFARM` is the first autoregressive model that can serve as an efficient framework for data acquisition.

## 4.3 Unconditional generation

`SFARM` is not only specially suited to conditional generation. We show that diverse generations can be achieved when sampling from scratch. In Figure 6, we include samples from three different image datasets.

Also, we include in Table 2 the resulting test negative log-likelihoods on MNIST and CIFAR-10. Whilst for MNIST, our NLL is comparable to the recent state of the art, for CIFAR-10, results are not that competitive.

Table 2: Test NLL results on binarized MNIST and CIFAR of our `SFARM` and recent state-of-the-art methods.

| Model | Test NLL |
|---|---|
| **Binarized MNIST** | (nats) |
| Locally Masked PixelCNN (Jain et al., 2020) | 77.58 |
| Efficient VDVAE (Hazami et al., 2022) | 79.09 |
| PixelRNN (Van den Oord et al., 2016c) | 79.20 |
| PixelCNN (Van den Oord et al., 2016c) | 81.30 |
| **SFARM (ours)** | 78.12 |

Table 3: Test NLL results on CIFAR-10 of our `SFARM` and recent state-of-the-art methods.

| Model | Test NLL | FID |
|---|---|---|
| **CIFAR-10** | (b/dim) | |
| IAF-VAE (Kingma et al., 2016) | $\leq 3.11$ | |
| BIVA (Maaløe et al., 2019) | $\leq 3.08$ | |
| NVAE (Vahdat & Kautz, 2020) | $\leq 2.91$ | |
| VDVAE (Child, 2020) | $\leq 2.87$ | |
| ARDM-Upscale 4 (Hoogeboom et al., 2022) | 2.64 | |
| PixelCNN++ (Salimans et al., 2017) | 2.92 | |
| DDPM (Ho et al., 2020) | 3.7 | 13.51 |
| Improved DDPM (Nichol & Dhariwal, 2021) | 2.94 | 3.27 |
| CTM (Kim et al., 2023) | 2.43 | 1.63 |
| FM $^w$/ OT (Lipman et al., 2022) | 2.99 | 6.35 |
| **SFARM (ours)** | 2.97 | 20.52 |

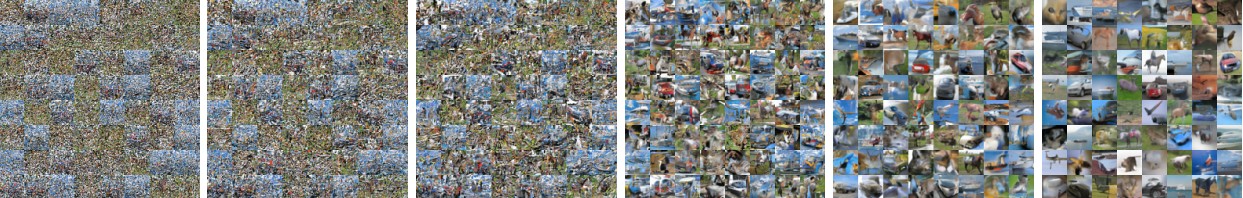

Figure 7: CIFAR-10 samples from `SFARM` after performing 2, 8, 32, 128 and 512 generation steps.

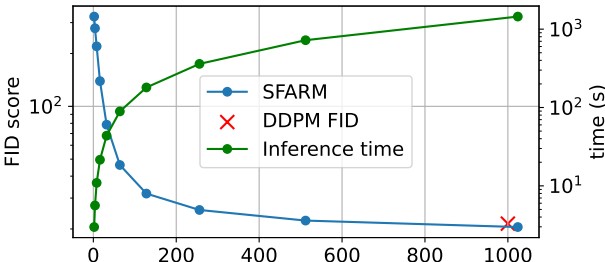

Figure 8: Evolution of the FID score of CIFAR-10 samples from `SFARM` over the number of generation steps.

### 4.3.1 Reducing sampling steps

One of the main limitations of `SFARM` is its sampling speed. To achieve the sharpest generations, it performs autoregressive steps, generating content element by element. However, due to its flexible design, `SFARM` can generate multiple elements simultaneously, as the trained network $p_\theta(\boldsymbol{x}_{\boldsymbol{z}_i}|\boldsymbol{x}_{\boldsymbol{z}<i})$ outputs the distribution of any unobserved element conditioned on arbitrarily observed elements.

A straightforward approach to improve sampling speed is to reduce the number of steps from $S$ to $S'$ by generating groups of elements at each step. In this approach, $i'$ is a downsampled version of the original $i$. Results following this strategy are provided in Figure 7. Low values of $S'$ produce decent results.

## 5 Limitations

Despite the promising results, the proposed family of models has some limitations. One notable limitation is that achieving high generation quality requires sampling fewer elements per step, which can slow down the generation process compared to one-step methods like VAEs. There are potential solutions for compacting information within the initial steps of the activation chain, which we plan to explore in future work.

## 6 Conclusion

In this paper, we introduced a novel family of deep generative models that leverage intrinsic data information and self-supervision principles, effectively generalizing previous order-agnostic models. By randomizing the per-element generation trajectory with a controlled stochastic approach, our models handle both structured and unstructured data types, demonstrating significant improvements in flexibility, simplicity, and accuracy. Our experiments, including image completion, missing data imputation and active information acquisition, show superiority or comparable performance in producing high-quality, realistic conditional samples even with minimal observed data.

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

# A  Experimental details

## A.1  Likelihood functions

The parametric families that we employ for the likelihood $p_\theta(\boldsymbol{x}_{\boldsymbol{z}_i}|\boldsymbol{x}_{\boldsymbol{z}_{i-1}})$ of different data types vary depending on their nature. We mainly employed Gaussian distributions for tabular datasets, Bernoulli distributions $\mathcal{B}(f_\theta(\boldsymbol{x}_{\boldsymbol{z}_{i-1}}))$ for binary images, and following Kingma et al. (2016), a mixture of discretized logistic distributions for coloured images, where both the locations $\boldsymbol{\mu}_l$, and scale $\boldsymbol{s}_l$ parameters are learned through a function $f_\theta(\boldsymbol{x}_{\boldsymbol{z}_{i-1}})$.

## A.2 Architecture details

Table 4 outlines the key hyperparameters used to reproduce our experiments. We use an input processor that expands the image channels to the desired `base_channels`, with one-quarter of these channels reserved for the class and activation mode embeddings. The `in_planes` parameter denotes the depth after the initial 1x1 convolution preceding the U-Net layers. The `layers` parameter refers to the number of symmetric downsampling and upsampling layers in the U-Net architecture. Each layer doubles the input depth while reducing the spatial dimensions by a factor of two. Additionally, we incorporate sinusoidal time embeddings into every U-Net block, adding them to the intermediate representations.

Following ARDMs, the smallest resolution processed by the U-Net is $32 \times 32$ (except for MNIST, where it remains at $28 \times 28$). This approach is typical for tasks involving NLL optimization and lossless compression, which often require high-resolution feature maps (Mentzer et al., 2019).

For tabular datasets, we utilize MLP networks with concatenated time embeddings. In this case, `units/layer` refers to the number of hidden units per layer in the MLP. "DL" stands for Discretized Logistic as defined by (Kingma et al., 2016), while "MoDL" refers to the Mixture of Discretized Logistics proposed by (Salimans et al., 2017). All models were trained using Nvidia Quadro RTX 5000 GPUs.

Table 4: Key hyperparameters for reproducing the experiments.

| Parameter | MNIST | Fashion-MNIST | CIFAR-10 | CelebA-HQ (64 x 64) | ImageNet (64 x 64) | Tabular datasets |
|---|---|---|---|---|---|---|
| base_channels | 32 | 32 | 256 | 256 | 256 | - |
| in_planes | 32 | 32 | 128 | 128 | 128 | - |
| layers | 1 | 1 | 1 | 2 | 2 | 3 |
| blocks/layer | 4 | 4 | 32 | 16 | 16 | - |
| units/layer | - | - | - | - | - | 32 |
| likelihood | Bernoulli | DL | MoDLs | MoDLs | MoDLs | Gaussian |
| n_mix | - | - | 30 | 30 | 30 | - |
| batch_size | 128 | 128 | 32* | 16* | 16* | 100 |
| epochs | 300 | 300 | 3000 | 3000 | 20 | 2K |
| lr | $2 \times 10^{-4}$ | $2 \times 10^{-4}$ | $2 \times 10^{-4}$ | $2 \times 10^{-4}$ | $2 \times 10^{-4}$ | $10^{-3}$ |

\* Parallelized over 4 GPUs.

# B Further theoretical analysis

## B.1 Marginal distribution of elements in the activation masks: the Uniform case

In this section, we aim to determine the parameters of the marginal distribution $p(z_{ij})$ when $p(\boldsymbol{z}_i^{\downarrow}|\boldsymbol{z}_{i-1})$ is defined as a Categorical distribution with uniform probabilities. Building on the discussions from Section 3.3, the conditional distribution for elements yet to be activated is given by the following Categorical distribution with uniform probabalities.

$$p(\boldsymbol{z}_i^{\downarrow}|\boldsymbol{z}_{i-1}) = \mathcal{C}\left(\frac{1}{S-i+1}\right). \tag{16}$$

For simplicity, we will omit the notation $\boldsymbol{z}_i^{\downarrow}$ in this context. The per-element probabilities can be expressed as

$$p(z_{ij} = 1|\boldsymbol{z}_{i-1}) = \frac{1}{S-i+1}. \tag{17}$$

By leveraging the total probability theorem, we can derive

$$p(z_{ij} = 1) = p(z_{i-1,j} = 0) \cdot p(z_{ij} = 1|z_{i-1,j} = 0) + p(z_{i-1,j} = 1) \cdot p(z_{ij} = 1|z_{i-1,j} = 1). \tag{18}$$

Letting $\alpha_i = p(z_{ij} = 1)$ and substituting the relevant conditional probabilities from Equation equation 16, we obtain

$$\alpha_i = (1 - \alpha_{i-1}) \cdot \frac{1}{S-i+1} + \alpha_{i-1} = \frac{1 + \alpha_{i-1}(S-i)}{S-i+1}. \tag{19}$$

Starting from $\alpha_0 = 0$, we calculate $\alpha_1 = \frac{1}{S}$. For $\alpha_2$, the calculation follows

$$\alpha_2 = \frac{1 + \alpha_1(S-2)}{S-1} = \frac{2(S-1)}{S(S-1)} = \frac{2}{S}. \tag{20}$$

Thus, the marginal probability of activation adheres to the formula

$$p(z_{ij} = 1) = \frac{i}{S}. \tag{21}$$

Or equivalently, the marginal probability of the element of a mask at step $i$ is given by

$$p(\boldsymbol{z}_i^{\downarrow}) = \mathcal{C}(i/S) \tag{22}$$

