# OpenReview forum: "Flexible Conditional Generation with Stochastically Factorized Autoregressive Models"
_TMLR — Withdrawn by Authors_

### Review · Reviewer_zUqr · 2025-02-05

**Summary Of Contributions:**

The authors introduce SFARM, a family of order-agnostic autoregressive models that leverages intrinsic data information and self-supervision principles. Their approach generalizes order-agnostic autoregressive models by defining a mixture of probabilistic activation mechanisms. The model was validated on restoration tasks, generation tasks and data ademonstrates superior performance in conditional generation tasks, and active feature acquisition.

**Audience:**

No

**Broader Impact Concerns:**

Not discussed.

**Claims And Evidence:**

No

**Requested Changes:**

Please see the weaknesses section.

**Strengths And Weaknesses:**

## Strengths

- Order-agnostic autoregressive model is an important topic in the literature of deep generative modeling.

---

## Weaknesses

**General comments**

- The abstract, introduction, and main body are not well aligned, making it difficult to grasp the paper’s scope, motivation, and contributions.
- Few recent papers from the past two years are cited, and no papers from 2024 are referenced.
- Section 2 does not sufficiently discuss previous works that are directly relevant to this study.
- Section 5 contains minimal discussion of the experimental results, making it difficult for readers to interpret the findings.

**The motivation of this paper is not very clear.**

- The relationshiop between "conditional generation" and "information acquisition" is not clear. In particular, although "information acquisition" is a key focus of this paper, the task itself is not explained.
- It is unclear why the authors propose a broader family of order-agnostic autoregressive models, as it does not seem directly relevant to their stated motivation.
- The active information acquisition component appears disconnected from the conditional generation aspect.
- What the authors express with "conditional generation" only becomes clear in the experimental section. They should better define their scope (perhaps "restoration" would be more appropriate) since conditional generation encompasses various tasks including class and text-conditional generation.
- Several claims lack proper support:
    - The statement *"deep autoregressive models are particularly well-suited for conditional generation tasks"* needs citations.
- The methodology section fails to explain how the proposed approach specifically addresses improvements in conditional generation and data acquisition tasks.


**The manuscript needs major revision to convey their contribution in a much easier way.**

- Many notations are not clearly explained. The followings are exmples.
    - $\boldsymbol{x}\_i$ is eplained as "intermediate stage of a generated data point". Is this different from $\boldsymbol{x}_{\boldsymbol{z}_i}$?
    - The mode parameter $m$, despite being a key contribution, is inadequately explained.
- Some notation choices are unnecessarily complex or non-standard (e.g., $(\boldsymbol{z}^{(a)},\boldsymbol{z}^{(b)})$, $(\boldsymbol{z}_i^\uparrow,\boldsymbol{z}_i^\downarrow)$)
- Section 3 is hard to follow.
    - The key technical differences and advantages of SFARM over ARDM are not well explained. Some figures/illustrations could help clarify these distinctions.
    - Figure 2, the only illustration in the methodology section, fails to effectively describe SFARM's key contributions.
    - The proposed loss function could be presented more concisely. Additionally, the validity of the inequality (5) is not well sepported. The reasoning *"our lower bound is equivalent to the standard ELBO used in VAEs if the variational posterior coincides with the prior"* is not convincing because such VAE's latent space is completely collapsed (a.k.a., posterior collapse)
    - Section 3.7 appears disconnected from the rest subsection in Section 3.


**The experiments and discussions are not very convincing.**

- The experimental setup lacks sufficient detail. The authors should provide more comprehensive explanations (possibly in an appendix) and/or include source code for reproducibility.

- The choice of VAE, MIWAE, and DDPM as baselines needs justification. Comparisons with discrete models (e.g., discrete diffusion models and MaskGIT) are missing.

- The image completion methodology for SFARM and baselines in Section 4.1 is unclear.

- The implementation of DDPM using an encoder-decoder without skip connections, rather than the standard UNet architecture, is not common and needs explanation.

- Section 4.2 lacks sufficient context about the data acquisition task, which should be better explained either in this section or in the related work.

- Section 4.3's discussion of test NLL results is too brief, failing to provide meaningful insights beyond stating SFARM's competitive performance (except for CIFAR10).

- (minor) I suspect inferece time is linear w.r.t. the nunmber of generation steps but the y axis of Figure 8 is log-scale.

---

### Review · Reviewer_4Kqf · 2025-03-05

**Summary Of Contributions:**

This paper proposes an autoregressive generative model with binary latent variables. The main motivation of the authors is designing a model which operates with a general dependency structure rather than a monotic one.

The authors provide experimental results on data imputation and on unconditional generation.

**Audience:**

Yes

**Broader Impact Concerns:**

I don't have particular concerns to report.

**Claims And Evidence:**

Yes

**Requested Changes:**

My main concern is clarity. The unconditional generation results, judging by the quantitative evaluation, seem to be a bit behind compared to the other methodologies.

I am not super upto date on the related literature, but it seems as if you are able to get good data imputation results. To make the paper more interesting and useful for people, I think section 3.2 and section 3.3. should be improved. It's really hard to understand what is the main contribution and the idea here. I suggest improving figure 2 to make the idea clearer and make it stand out. Right now, I can not understand what is being proposed such that you can learn a more complex dependency structures. For this I think section 3.3 can use a lot of improvement also. It's very hard to pick out the main idea at the moment.

**Strengths And Weaknesses:**

Strengths:
- The authors have a clear goal of developing an autoregresive model that figures out complex dependencies

- The experiments on images indicate that the proposed model performs well for data imputation.

Weaknesses:
- I find that the paper, in the methodological section is not very clear. I am having hard time understanding the core idea, and how you are able to figure out complex dependencies. I would perhaps suggest to improve figure 2, to convey the idea in a more clear and intuitive way.

- What exactly makes it so that you can figure out more general dependency structures? Do you have a model such as an RNN that selectively gathers earlier examples?

- I feel like the methods you are comparing against in Table 3 are a bit dated, and still the performance seems to be behind those.

- For unconditional generation on cifar-10, as also admitted by the authors, the generation performance is not competitive. It remains a concern as to whether the proposed method would be competitive on more complex and higher resolution images.

---

### Review · Reviewer_BpKv · 2025-04-15

**Summary Of Contributions:**

This paper presents an autoregressive generative model for conditional generation, denoted as SFARM. The main feature of the method is that it allows for a stochastic schedule of the intermediate steps of the generation process, as oppose to following a deterministic sequence. For example, for image generation, instead of following a pre-defined schedule for generating the pixels such as from left to right and top to bottom, SFARM is trained with masks of pixels sampled uniformly. The goal and intuition is that the stochastic schedule used during training allows for more arbitrary conditions (for example, a random sample of pixels, or other arbitrary masks) at generation time.

**Audience:**

Yes

**Broader Impact Concerns:**

In my opinion, the use of facial images data needs to be strongly justified and discussed.

**Claims And Evidence:**

No

**Requested Changes:**

In the previous section I have offered a discussion of the aspects that in my opinion could be improved and adjusted. As a summary:

- Motivation:
    - Clarify the needs and gaps in the literature that the paper aims to address
    - Situate the paper in the literature without misrepresenting existing methods
- Soundness
    - Adjust the claims of novelty
    - Adjust the claims of superiority and effectiveness compared to other methods
    - Improve the evaluation of the model by selecting competitive and suitable baselines
    - Demonstrate the effectiveness of the method in relation to the stated goals and motivation
- Clarity
    - Improve the structure and flow of the abstract and introduction
    - Adjust the rigour in the notation and definition of the mathematical descriptions
    - Improve the structure of the results section

**Strengths And Weaknesses:**

As strengths of the paper, I would point out that the technical part of the paper where the method is described (Section 3) provides the context, background and notation to introduce the details. Despite some aspects that could be improved about the clarity (see below), for the most part it is easy to follow the description of the method and the figures, algorithms and mathematical descriptions support the text. Furthermore, I particularly appreciate that the authors include a section to discuss the limitations of the method.

Nonetheless, I have several major concerns regarding the soundness and clarity of the paper, on which I will elaborate below.

### Soundness

In my opinion, there are a number of aspects where the scientific quality of the paper could be substantially improved.

#### Motivation

Starting with the introduction, I would point out that the motivation for the method relies on unsubstantiated claims. For example, the first paragraph attempts, as far as I understand, to motivate the need for a new conditional generative model, by claiming that existing methods fail or are not suitable for this purpose. However, these claims are vague and not convincing. By way of illustration, I will quote the argument about adversarial methods:

> Adversarial methods like Generative Adversarial Networks (GANs, Goodfellow et al., 2014) provide powerful capabilities for generating high-quality images but are notoriously difficult to train and, in their basic form, do not naturally support conditional generation.

First, the authors restrict the large family of adversarial methods to GANs and merely cite the original 2014 paper, ignoring the long and broad literature on conditional adversarial methods, dating as far back as the work by Mirza and Osindero in 2014. Conversely, the papers praises the suitability of deep autoregressive models for conditional generation by merely mentioning "their inherent design and straightforward learning process via maximum likelihood estimation". Beyond these points, I would argue that the motivation for the method, the need for this kind of generative model and the gaps that this method specifically addresses is rather weak.

#### Claims and evidence

Perhaps a more concerning weakness for me is my assessment that the claims of contributions and the stated goals are hardly supported by the methods and results. The first stated contribution is that the paper presents "a new family of order-agnostic autoregressive models" (Introduction) and "a novel family of deep generative models" (Conclusion). In my humble opinion, I would argue that these claims overstate the methodological contribution of the paper. I ignore whether other methods in the literature have proposed similar autoregressive generative models with stochastic masks, but even granting that this is a novel method, it would be a rather limited adjustment of existing methods. This by itself is not a weakness, as I strongly hold that methodological novelty is orthogonal to significance or usefulness. However, I think it is fair to say that the presented method does not represent a novel family of generative models.

Therefore, it is important for me to assess whether the paper "demonstrate[s] the effectiveness of [the] proposed method" and the "the superiority of [the] proposed family of models over alternatives in relevant downstream tasks" (Introduction). To this end, I can comment on the experimental setup and evaluation of the model (Section 4).

First of all, the choice of baselines is arguably not representative of the state of the art on the evaluated tasks. These baselines are the VAE by Kingma and Welling (2013), that is 12 years old, a variation of MIWAE (2016) from 2019 and the RePaint variation from 2022 of denoising diffusion (2020). Second, I fail to understand the relevance of the tasks selected for the evaluation of a conditional generative model. They seem akin to the typical tasks tackled by in-painting methods, where parts of the inputs (images or tables) are masked. In this regard, I would expect state-of-the-art in-painting methods to be more suitable baselines for comparison. Third, the evaluation procedure could be stronger. There is no discussion of the metrics, the evaluation is mostly qualitative, showing a few images that could be cherry-picked and again there are some claims that are not supported by the results offered in the paper. For example, it is claimed that "[i]t is evident that foundational VAEs do not offer an accurate solution for dynamically acquiring new variables", referring to the results in Figure 5. However, the VAE results in Figure 5b are equivalent to those of SFARM and it is unclear whether the results in 5a and 5c could be matched with better model selection. Finally, the set of results provided in the paper are nevertheless not convincing to me that the proposed method is effective and superior to alternative methods. I find it surprising and telling that as motivation of the method, the authors remark that "PixelCNN and its variants struggled to generate diverse and realistic samples from complex datasets", but the experimental setup of the paper is focused on data sets such as Fashion-MNIST, CIFAR-10 and Celeba.

Finally, in Section 3, there are several claims praising the benefits of the proposed method that I do not think are validated or confirmed by the results. For example: 1) that the method does not require an additional regularization term, simplifying the training process while maintaining robust and effective performance; 2) that it offers greater flexibility compared to alternative methods such as ARDMs.

#### Clarity

There are several aspects regarding the clarity of the paper that can be largely improved in my opinion. First and foremost, I found the abstract and the introduction quite hard to follow. In other words, it was not straightforward for me to understand the motivation, goals and object of the paper from the text. My assessment is that the text does not follow a clear logical structure and instead alternates between specific and general aspects. A reference that I find helpful and easy to read about scientific writing is the paper by Mensh and Kording (2017) [_Ten simple rules for structuring papers_](https://journals.plos.org/ploscompbiol/article?id=10.1371/journal.pcbi.1005619).

Section 3 is easier to follow, as I mentioned above, but I would like to point out a few aspects that could be improved, in my opinion. The most important point to me is to improve the rigour regarding the mathematical notation and definitions. I have noticed that not all mathematical elements are defined before or immediately after they are introduced. For example, $S$ is used in Equation 1 but is not defined until three paragraphs later. Similarly, $k$, $m$ and $M$ are not defined in Equation 2. As another example, Equation 8 redefines $\mathcal{L}$, causing confusion.

Finally, another aspect that undermines the clarity of the paper is the lack of details about the experimental setup and how the proposed model and the baselines are trained. Some details are offered in the appendix but in my opinion it is important that the fundamental details are provided in the main paper. Otherwise, the paper becomes less clear and the results less convincing. I would also suggest to envision a consistent structure of Section 4, where the experiments are well justified and explained and the analysis is consistent. For example, I find it confusing that the results of some data sets (ImageNet) are not included in the paper despite being mentioned at the beginning. It is also very important in my opinion to offer an in-depth analysis of the results, which is currently lacking.

### References

- Mirza, M., & Osindero, S. (2014). Conditional generative adversarial nets. NeurIPS.
- Mensh, B., & Kording, K. (2017). Ten simple rules for structuring papers. PLOS Computational Biology.

---

### Note · Authors · 2025-04-28

**Comment:**

After carefully considering the reviewers’ feedback, we believe that addressing all the comments would require substantial changes, effectively resulting in a completely new version of the paper. Therefore, we have decided to withdraw the current version and plan to submit a significantly improved manuscript in the future. We thank the reviewers and the editorial team for their time and consideration.

**Withdrawal Confirmation:**

I have read and agree with the venue's withdrawal policy on behalf of myself and my co-authors.